

# Green manure combined with reduced nitrogen reduce NH₃ emissions, improves yield and nitrogen use efficiencies of rice

Zhongze Hu[1], Daliu Yang[1], Yaming Feng[1], Shuanglin Zhang[1], An Wang[1], Qiaozhen Wang[2], Yayun Yang[2], Chunying Chen[2], Yuefang Zhang[3] and Xian Wang[1]

[1] Institute of Taizhou Agricultural Science, Jiangsu Academy of Agricultural Sciences, Taizhou, Jiangsu, China
[2] Agro-Tech Extension and Service Center of Hailing District, Taizhou, Jiangsu, China
[3] Institute of Agricultural Resources and Environment, Jiangsu Academy of Agricultural Sciences, Nanjing, Jiangsu, China

## ABSTRACT

**Background**. Green manure is an important source of organic fertilizer. Exploring green fertilizer and nitrogen fertilizer reduction is important for agricultural production. However, few studies have been conducted, especially on the effects of different green fertilizers along with reduced nitrogen fertilizer application on soil ammonia volatilization emissions, rice yield, and nitrogen fertilizer uptake and utilization.

**Methods**. In this study, the effects of different types of green manure and reduced nitrogen fertilizer application on soil ammonia volatilization emissions, aboveground population characteristics of rice, and nitrogen fertilizer uptake and utilization were explored. This study was based on a field-positioning experiment conducted between 2020 and 2022. Six treatments were established: no nitrogen fertilizer application (CK), conventional fertilization in wheat-rice (WR), villous villosa-rice (VvR), vetch sativa-rice (VsR), rapeseed seed-rice (RR), and milk vetch-rice (GR), with a 20% reduction in nitrogen fertilizer application. The amounts of phosphorus and potassium fertilizers remained unchanged. The characteristics of ammonia volatilization loss in rice fields, agronomic traits of rice, yield traits, and nitrogen uptake and utilization were investigated.

**Results**. The results indicated a significant difference ($P < 0.05$) in the impact of different treatments on ammonia volatilization emissions from rice in the two-year experiment. Compared with WR treatment, VvR, VsR, RR, and GR treatments reduced the total ammonia volatilization loss by 23.58 to 39.21 kg ha$^{-1}$, respectively. Compared with the conventional WR treatment, other treatments increased rice yield by 0.09 to 0.83 t ha$^{-1}$. GR treatment was significantly higher than other green fertilizer treatments, except for VsR ($P < 0.05$). It increased the nitrogen uptake of rice by an average of 4.24%–22.24% and 13.08%–33.21% over the two years, respectively. The impact of different types of green manure on the nitrogen uptake and utilization of rice varied greatly, indicating that the combination of green manure and fertilizer is a sustainable fertilization model for crops to achieve high yields. In particular, the Chinese milk vetch as green manure was more beneficial for ammonia volatilization reduction in paddy field and stable grain production of rice.

Corresponding author
Xian Wang, 43850149@qq.com

## INTRODUCTION

Rice (*Oryza sativa* L.) is one of the most important food crops worldwide, accounting for nearly half of the global population (*Shi et al., 2023*). The increase in population has led to a higher demand for food production. The widespread application of chemical fertilizers is an important measure for increasing rice yield (*Yang et al., 2020*). However, excessive fertilization not only causes crop yield reduction but also causes environmental pollution, such as low fertilizer utilization, soil degradation, and eutrophication of water bodies due to excessive ammonia volatilization (*Qiu et al., 2022*; *Wang et al., 2023a*; *Wang et al., 2023b*; *Wang et al., 2023c*). Nitrogen loss through the volatilization of ammonia accounts for 29–35% of the total nitrogen application (*Xu et al., 2017*). Therefore, it is extremely important to study measures to reduce ammonia volatilization in farmlands, including increasing crop yields, improving nitrogen utilization efficiency, and enhancing the ecological environment.

Ammonia volatilization in farmlands is influenced by factors such as fertilizer dosage and type, straw return to the field, soil properties, and climatic conditions (*Sha et al., 2019*; *She et al., 2018*). Many technical measures have been developed to reduce ammonia volatilization, such as improving fertilization techniques (*e.g.*, deep placement of N fertilizer or split applications of N fertilizer) (*Zhang et al., 2022*; *Feng et al., 2017*), application of slow/controlled-release fertilizers or urea combined with urease inhibitors (*Zhou et al., 2023*; *Liu et al., 2023*; *Hu et al., 2023*), and the use of new types of biochar (*Chu et al., 2020*). However, efficiency-enhancing fertilizers and biochar are often too expensive and the technology is too complex, which limits their use in rice production. Research has shown that the amount of nitrogen fertilizer used is the main factor affecting ammonia volatilization (*He et al., 2014*). As the nitrogen input increases, ammonia volatilization losses continue to intensify. The combination of nitrogen reduction and deep fertilization can further reduce ammonia volatilization emissions. As nitrogen input increases, ammonia volatilization emissions from farmlands also increase (*He et al., 2014*). *Ma et al. (2023)* observed that using agricultural waste as a membrane material, in addition to reducing nitrogen fertilizer reduced the total ammonia volatilization emissions during rice cultivation by 19% to 31%. This indicates that optimizing fertilization can significantly reduce soil ammonia volatilization loss. Reasonable nitrogen fertilizer application can ensure stable crop yield while also considering the ecological environment.

Green manure, a nutrient-rich biological fertilizer that uses plant nutrients to replenish the soil, plays an important role in increasing crop yield, enhancing soil fertility, improving soil physicochemical properties, regulating farmland ecology, and maintaining biodiversity (*Liang et al., 2022*). *Sharma et al. (2022)* confirmed that green manure can regulate soil physical and chemical properties after a five-year field experiment. Green manure is used as an environment-friendly technology to enhance paddy production in Iran. This practice

not only helps reduce environmental and health concerns but can also increase productivity (*Valizadeh et al., 2023*). *Islam et al. (2019)* found that the combination of green manure with nitrogen fertilizer consistently improved important morpho-physiological traits, such as chlorophyll content (SPAD value) and leaf area index (LAI), potentially leading to higher grain yield. Studies have shown that the combined application of Chinese milk vetch and rice straw can enhance soil fertility, reduce chemical fertilizer usage, and improve fertilizer conservation and efficiency (*Liang et al., 2022*; *Wang et al., 2022a*; *Wang et al., 2022b*). Returning straw from green manure to the field can increase soil organic matter and the decomposition of organic matter can produce organic acids. The humus formed by it can enhance the ability of the soil to adsorb $NH_4^+$ while also reducing the pH value of paddy soil and inhibiting soil ammonia volatilization (*Liang et al., 2022*).

Green manure is a crop mixture that includes leguminous crops with a biological nitrogen fixation function, mainly composed of Chinese milk vetch and wild pea, and non-leguminous plants, such as rapeseed seeds (*Valizadeh et al., 2023*). Chinese milk vetch is a high-quality winter-fallow green fertilizer for rice fields, and its root system can enhance soil nitrogen levels through symbiotic nitrogen fixation. Planting Chinese milk vetch in winter can effectively control weeds in farmland and cover exposed ground, reducing the ammonia volatilization flux in rice fields (*Zheng et al., 2023*; *Ma et al., 2020*). The return of Chinese milk vetch to the field can also provide abundant carbon and nitrogen sources for soil microorganisms, thereby promoting the conversion of inorganic nitrogen into organic nitrogen in the soil. This process helps decrease the levels of inorganic nitrogen in the soil and ultimately reduces the overall emission of ammonia volatilization (*Wang et al., 2023a*; *Wang et al., 2023b*; *Wang et al., 2023c*; *Liu et al., 2020*). Studies have shown that returning Chinese milk vetch to the soil can partially substitute for chemical fertilizers, enhance soil organic matter accumulation, decrease ammonia volatilization emissions, improve crop nitrogen utilization efficiency, and increase crop yield and economic benefits (*Zhang et al., 2020*; *Xie et al., 2016*). Returning rapeseed seeds as a green fertilizer to the soil can enhance peanut growth by increasing the number of effective root nodules and individual plant dry weight of subsequent peanut crops (*Zheng et al., 2019*). It can also improve the soil structure, activate insoluble phosphorus in the soil, enhance soil nutrient content, increase soil microbial metabolic activity, promote nutrient uptake and utilization by subsequent rice crops, and increase yield (*Liang et al., 2022*; *Wang et al., 2022a*; *Wang et al., 2022b*). Although there have been some studies on the effects of returning green manure on soil ammonia volatilization emissions, rice growth, and yield, it remains unclear which green manure application is most beneficial for reducing ammonia volatilization and ensuring stable grain production. Therefore, this experiment used conventional rice and wheat rotation as a control to comprehensively evaluate the effects of four green fertilizers, namely milk vetch, rapeseed seed, vetch pea, and villous pea, on ammonia volatilization, rice growth, and yield in rice fields, with the aim of providing technical support for green rice production.

**Table 1  Basic physical and chemical properties of soil before the experiment from 2020 to 2022.**

| pH | Organic matter g kg$^{-1}$ | Total nitrogen (N) g kg$^{-1}$ | Available N mg kg$^{-1}$ | Available P mg P$_2$O$_5$ kg$^{-1}$ | Available K mg K$_2$O kg$^{-1}$ |
|---|---|---|---|---|---|
| 6.6 | 32.7 | 1.9 | 110.7 | 8.3 | 102.8 |

## MATERIALS & METHODS

### Materials and study site

The experiment started in October 2020, and the soil tested was heavy loamy black clay at the Taizhou Institute of Agricultural Sciences, Jiangsu Academy of Agricultural Sciences (119°59′38″E, 32°32′23″N). The physical and chemical properties of the soil before the green fertilizer experiment are listed in Table 1. Wheat (*Triticum aestivum* L.), rice (*Oryza sativa* L.), villous villosa (*Vicia villosa* L.), vetch sativa (*Vicia sativa* L.), rapeseed (*Brassica campestris* L.), and Chinese milk vetch (*Astragalus sinicus* L.) were provided by Jiangsu Suzhong Agricultural Development Co., Ltd. The experimental area has an average elevation of two meters and a subtropical monsoon climate. The average annual precipitation of the experiment is 1051 mm, the temperature is 14.8 °C, the annual sunshine duration is 2233 h, and the frost-free period is 218 days. The weather conditions during the experiment are shown in Fig. 1.

### Experimental design

Six treatments were set up in the experiment: wheat rice without nitrogen fertilizer (CK), wheat-rice conventional fertilization (WR), villous villosa-rice (VvR), vetch sativa-rice (VsR), rapeseed seed-rice (RR), and milk vetch-rice (GR), with a plot area of 30 m$^2$ and three replicates. The CK treatment did not involve the use of nitrogen fertilizer, whereas the WR treatment used 270 kg ha$^{-1}$ nitrogen, and the other four treatments used 216 kg ha$^{-1}$ nitrogen (20% reduction in nitrogen) (*Pu et al., 2023*). During the basal fertilization period, which coincided with the date of transplantation, 40% of the applied chemical N was evenly distributed in the surface water for each fertilizer treatment. Subsequently, during the tillering fertilization period (eight days after transplantation), 30% of the chemical N was broadcast. Finally, during the panicle fertilization period (40 days after transplantation), the remaining 30% of the chemical N was broadcast. Phosphorus fertilizer (75 kg P$_2$O$_5$ ha$^{-1}$) was applied all at once as base fertilizer, whereas potassium fertilizer (75 K$_2$O ha$^{-1}$) was applied in a 1:1 ratio for both base and ear fertilizers. The previous wheat straw was returned to the field with WR and CK treatments at the mature stage, whereas the green manure was returned to the field with GR, VvR, RR, and VsR treatments at the flowering stage. The nitrogen nutrient inputs for the different rice seasons are listed in Table 2. The nitrogen, phosphorus, and potassium fertilizers used in the experiment were urea (N, 46%), superphosphate (P$_2$O$_5$, 52%), and potassium sulfate (K$_2$O, 12%), respectively. Other agricultural operations were consistent and implemented uniformly according to local production.

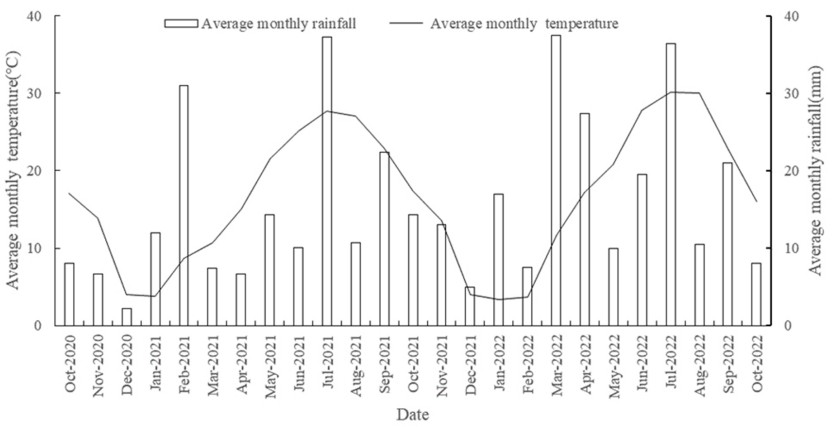

**Figure 1** **Monthly average temperature and precipitation at experimental site from October 2020 to October 2022.**

**Table 2** **Nitrogen input for each treatment.** Wheat rice without nitrogen fertilizer (CK), wheat-rice conventional fertilization (WR), villous villosa-rice (VvR), vetch sativa-rice (VsR), rapeseed-rice (RR), and milkvetch-rice (GR). Different lowercase letters indicate that the differences reached a significant level of 0.05.

| Year | Treatment | Chemical nitrogen input (kg ha$^{-1}$) | Total nitrogen content in aboveground parts of plants (kg ha$^{-1}$) | Total nitrogen content of plant roots (kg ha$^{-1}$) | Total nitrogen content of plants (kg ha$^{-1}$) | Total nitrogen input (kg ha$^{-1}$) |
|---|---|---|---|---|---|---|
| 2021 | WR | 270 | 23.16 d (Wheat) | 1.36 e | 24.52 d | 294.52 |
| | VvR | 216 | 39.54 c (Vicia villosa) | 8.45 c | 47.99 b | 263.99 |
| | VsR | 216 | 59.04 b (Vicia sativa) | 7.19 a | 66.23 a | 282.23 |
| | RR | 216 | 26.37 d (Rapeseed) | 3.34 b | 29.72 c | 245.72 |
| | GR | 216 | 69.00 a (milk vetch) | 10.60 d | 79.60 a | 295.60 |
| | CK | 0 | 22.73 d (Wheat) | 1.24 e | 23.97 d | 23.97 |
| 2022 | WR | 270 | 20.63 c (Wheat) | 1.46 c | 22.09 d | 292.09 |
| | VvR | 216 | 45.31 b (Vicia villosa) | 9.42 a | 54.73 b | 270.73 |
| | VsR | 216 | 46.73 b (Vicia sativa) | 10.35 a | 57.08 b | 273.08 |
| | RR | 216 | 25.79 c (Rapeseed) | 4.35 b | 30.14 c | 246.14 |
| | GR | 216 | 73.64 a (milk vetch) | 11.24 a | 84.88 a | 300.88 |
| | CK | 0 | 21.44 c (Wheat) | 1.38 c | 22.82 d | 22.82 |

## Sampling and measurements
### Ammonia volatilization

The intermittent closed-chamber air extraction method was used to collect volatilized ammonia from paddy soil (*Zhao et al., 2011*). The collection device consisted of a vacuum pump, a sealed chamber, a gas cylinder, and a throttle valve. The enclosed room was a cylindrical chamber made of organic glass with an open bottom, inner diameter of 14 cm, and height of 15 cm. We embedded a sealed chamber in the surface soil between the two rows of transplanted rice, leaving a space of 8–10 cm in height inside. There were two holes

on the top of the cover, one of which was a 25 mm diameter air inlet. The air inlet was connected to a height of 2.5 m to minimize the impact of air exchange on the measurement of ammonia volatilization in the rice fields. The other was a gas collection port connected to an absorption bottle containing 2% boric acid, which was then connected to a vacuum pump. Ammonia volatilization was achieved by setting the indoor air exchange rate to 15–20 times per minute using a flow meter. The pumping period was in the morning (8:00–10:00) and afternoon (14:00–16:00) during the rice season. After sampling was completed, the sample was taken to the laboratory for titration with dilute sulfuric acid, and the daily ammonia volatilization flux was calculated.

## SPAD of leaf

Twenty days after rice heading, the chlorophyll content in the upper, middle, and lower leaves of the main stem was measured using a chlorophyll meter SPAD-502 Plus (Konica Minolta, Shanghai, China).

## Crop yield and nitrogen content

After the wheat had matured, the grains were collected for yield and total nitrogen content measurements, and the remaining straw was returned to the field. The dry matter weight and total nitrogen content of the green manure crops were measured during the full flowering period. The remaining straw was then turned over and returned to the field. At the maturity stage of rice, a 2 $m^2$ plot with uniform growth in each plot was selected. The number of panicles per unit area, number of grains per panicle, seed-setting rate, and 1000-grain weight with full grains were calculated. Furthermore, we calculated the rice grain yield (with 14% water content) and dried and weighed the straw to determine the straw yield. The total nitrogen content of the straw and grain was measured through a company (Nanjing Ruiyuan Biotechnology Co., Ltd. Nanjing, China).

## Statistical analysis and count

①Daily flux of ammonia volatilization $(kghm^{-2} d^{-1})$) = (concentration of $NH_-^{+4}$ N in the uptake solution $(kg L^{-1})$ × volume of dilute sulfuric acid uptake solution (L) × $10^{-6}$)/(exchange room area$(hm^{-2})$ × collection time (d) × $10^{-4}$).

②Ammonia volatilization loss rate (%) = ammonia volatilization flux (kg $hm^{-2}$)/chemical nitrogen fertilizer application rate (kg $hm^{-2}$).

③Emission coefficient of ammonia volatilization (%) = (ammonia volatilization in the nitrogen application area (kg $hm^{-2}$) − ammonia volatilization in non-nitrogen application areas (kg $hm^{-2}$))/nitrogen rate (kg $hm^{-2}$) × 100.

④Emission intensity of ammonia volatilization $(kgt^{-1})$ = total ammonia volatilization emissions per unit area $(kghm^{-2})$/rice grain yield per unit area (t $hm^{-2}$).

⑤Plant nitrogen accumulation (kg) = nitrogen content of straw (g $kg^{-1}$) × straw yield per unit area (t) +nitrogen content of grain (g $kg^{-1}$) × grain yield per unit area (t).

⑥Nitrogen harvest index (%) = grain nitrogen accumulation (kg) / Plant nitrogen accumulation (kg) × 100

⑦Nitrogen use efficiency (%) = (nitrogen accumulation of plants in nitrogen application areas (kg) − nitrogen accumulation of plants in non-nitrogen application areas (kg)/nitrogen rate (kg) × 100.

All data were analyzed using GraphPad Prism (version 8.0.0; GraphPad Software, San Diego, CA, USA, http://www.graphpad.com). Differences between groups were analyzed using one-way analysis of variance (ANOVA), followed by Dunnett multiple comparisons. Statistical significance was set at $P<0.05$.

## RESULTS

### Nitrogen input during the rice season

The analysis of nitrogen input before the rice growth season in 2021 and 2022 (Table 2) showed that there were significant differences in nitrogen nutrient content between the aboveground and root parts of the crops under different treatments ($P<0.05$). There were no significant differences in the nitrogen nutrient content between the aboveground and root systems of the crops between years ($P<0.05$). The GR treatment exhibited the highest nitrogen content in the aboveground and root systems of the different treatments in the two-year experiment, with 71.32 and 10.92 kg ha$^{-1}$, respectively. These values were significantly higher than those of the other treatments ($P<0.05$). The difference in total nitrogen input among the different treatments was GR>WR>V sR>VvR>RR.

### Dynamic changes in ammonia volatilization

The daily flux of ammonia volatilization in paddy soils in 2021 and 2022 is shown in Fig. 2. During the study period, after fertilization, the daily flux of ammonia volatilization in the soil exhibited a rapid increase, reaching its peak on the second day and subsequently decreasing. After monitoring for 10–15 days, there was no significant difference in the daily flux of ammonia volatilization among the different treatments. However, there were significant differences in the daily flux of ammonia volatilization in paddy soils between years ($P<0.05$).

During the base fertilizer period of the 2021 rice season, the daily ammonia volatilization flux of the different soil treatments reached a peak of 0.29–7.47 kg hm$^{-2}$ d$^{-1}$ on the second day of monitoring. ANOVA showed that the peak daily ammonia volatilization flux was significantly higher than the VsR and RR ($P<0.05$), and there was no significant difference between the VvR and GR. Furthermore, there were no significant differences between the VvR, VsR, GR, and RR. During the tillering stage, the daily flux of ammonia volatilization in different soil treatments reached a peak of 0.40–6.22 kg ha$^{-1}$ d$^{-1}$ on the second day of monitoring. A significant difference was observed between WR and VvR, GR, VsR, and RR ($P<0.05$). However, there were no significant differences among GR, VsR, VvR, and RR treatments. During the panicle stage, the daily ammonia volatilization flux of the different soil treatments reached a peak of 0.29–8.64 kg ha$^{-1}$ d$^{-1}$ on the second day of monitoring. The difference in the daily ammonia volatilization flux among the different soil treatments was consistent with that observed during the tillering stage.

During the base fertilizer period of the rice season in 2022, the daily ammonia volatilization flux of each treatment reached a peak of 0.26–7.96 kg ha$^{-1}$ d$^{-1}$

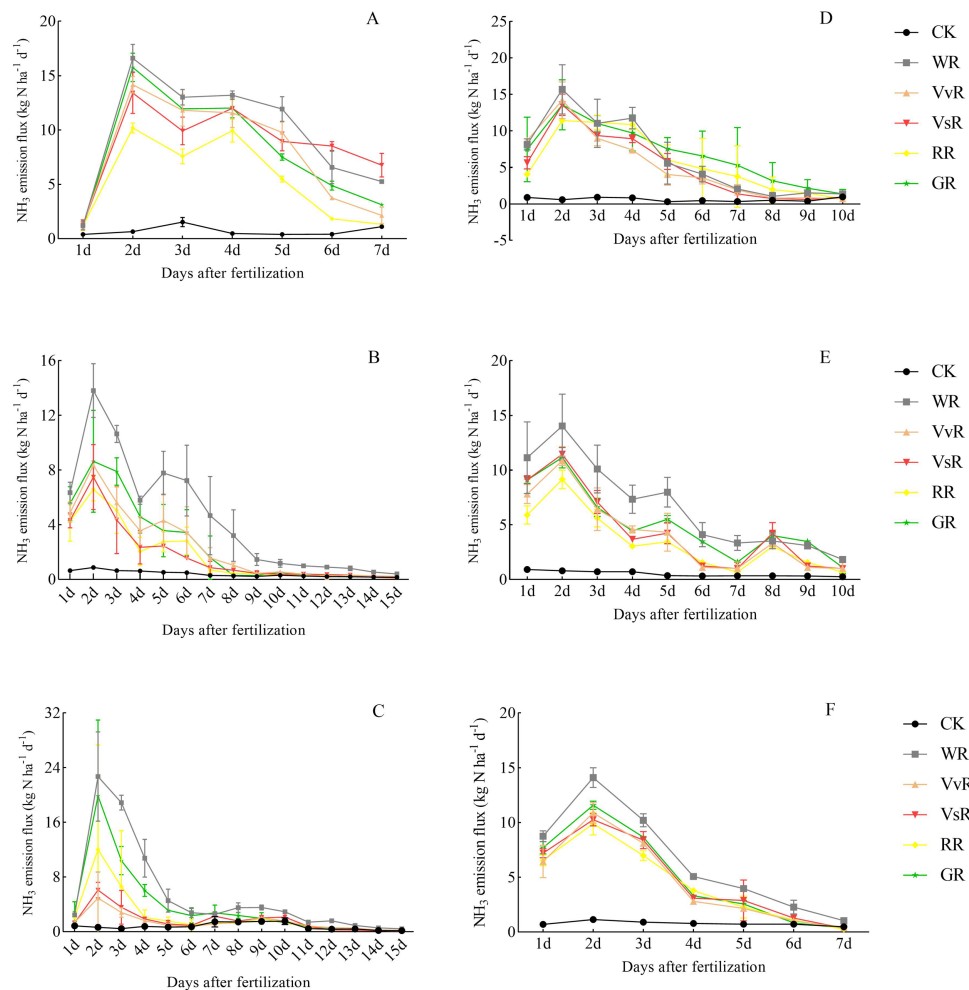

**Figure 2 Daily flux change of ammonia volatilization in paddy fields after fertilization in 2021 and 2022.** The vertical bars represent the standard deviation of the mean ($n = 3$). Abbreviations are the same as those in Table 2. The daily fluxes of ammonia volatilization from paddy fields after fertilization at basal (A), tillering (B) and panicle (C) stages in 2021 and basal (D), tillering (E) and panicle (F) stages in 2022.

on the second day of monitoring. The differences among the treatments were WR>GR>VvR>VsR>RR>CK. The daily flux of ammonia volatilization in WR soil was significantly higher than that in VvR, GR, VsR, and RR soils ($P<0.05$). The daily flux of ammonia volatilization in the GR, VsR, and VvR soils was significantly higher than that in the RR ($P<0.05$). During the tillering stage, the daily flux of ammonia volatilization in different soil treatments reached a peak of 0.36–6.31 kg $hm^{-2}$ $d^{-1}$ on the second day of monitoring. ANOVA revealed that the peak daily ammonia volatilization flux WR was not significantly different from the VvR, GR, and VsR treatments but was significantly different from RR ($P<0.05$). Differences in GR, VsR, VvR, and RR were not significant. At the heading stage, the daily ammonia volatilization flux of soil under different treatments reached a peak on the second day of monitoring, ranging from 0.52 to 6.34 kg $ha^{-1}d^{-1}$. The differences among the treatments were WR>GR>RR>VsR>VvR>CK. ANOVA of the

peak daily ammonia volatilization flux showed significant differences ($P<0.05$) between WR and VvR, GR, VsR, and RR treatments. No significant differences in GR, VsR, VvR, or RR were observed.

## Ammonia volatilization loss and loss rate

The emission and loss rates of ammonia volatilization from rice fields under different treatments over the two years are shown in Table 3. After applying the base fertilizer, there was a significant difference in ammonia volatilization emissions among the different treatments ($P<0.05$), with an average of 16.68–30.17 kg ha$^{-1}$. There were no significant differences in nitrogen content between the aboveground and root systems of the crops between years ($P<0.05$), followed by WR, VsR, GR, VvR, and RR, in descending order. Compared with WR, the VvR, VsR, RR, and GR averages reduced ammonia volatilization emissions from base fertilizers by 19.02–44.71% over the two years. After applying tillering fertilizer and ear fertilizer, the average ammonia volatilization emissions of different fertilization treatments for two years were between 13.52–28.94 kg ha$^{-1}$ and 12.36–22.66 kg ha$^{-1}$, respectively. The differences between the treatments were consistent with those of the base fertilizer. Compared with WR, VvR, VsR, RR, and GR reduced the total ammonia volatilization loss during the rice season by 37.02%, 31.97%, 47.68%, and 28.56%, respectively. The proportions of ammonia emissions from basal, tillering, and panicle fertilizers during the total growth period under different nitrogen application treatments were 37.0–45.44%, 29.08–33.64%, and 24.25–29.08%, respectively. Overall, the amount of ammonia volatilization emissions was in the order basal>panicle>tillering fertilizers. The effects of different treatments on the rate of ammonia volatilization loss in rice differed significantly ($P<0.05$; Table 3). After applying the base fertilizer, the average ammonia volatilization loss rate of each treatment over two years ranged from 25.74% to 37.75%. The RR treatment had the lowest loss rate, which was significantly lower than that of the other treatments ($P<0.05$). After tillering and ear fertilizer application, the ammonia volatilization loss rates in each treatment were 20.86–35.73% and 14.31–20.98%, respectively. The ammonia volatilization loss rate of RR during the entire growth period was the lowest, at 19.70%. Compared with WR, the ammonia volatilization loss rates of the VvR, VsR, RR, and GR treatments decreased by 21.72%, 14.96%, 34.94%, and 10.70%, respectively.

## Rice yield and ammonia volatilization emission intensity

ANOVA revealed that the impact of the year on seed setting rate and grain yield exhibited highly significant differences ($P<0.01$) but had no significant impact on rice straw yield and grain composition factors (Table 4). The treatment had no significant effect on seed setting rate but had a significant impact on rice straw yield, grain yield, and their constituent factors ($P<0.05$). The interaction between the year and treatment had a highly significant impact on grain and straw yield ($P<0.01$) but had no significant impact on grain composition.

Further analysis of rice yields revealed that GR grain and straw yields were the highest among the different treatments in both seasons. The grain yield of GR in 2021 was 10.71 t ha$^{-1}$, which was significantly higher than that of the other treatments, except for VsR

Hu et al. (2024), *PeerJ*, DOI 10.7717/peerj.17761

**Table 3  Effects of different treatments on ammonia volatilization loss and loss rate in paddy fields.** "-" means no data. Different lowercase letters indicate that the differences reached a significance level of 0.05. Abbreviations are the same as those in Table 2.

| Year | Treatment | Base fertilizer | | | Tillering fertilizer | | | Panicle fertilizer | | | Total cumulative emission of ammonia volatilization (kg ha$^{-1}$) | Ammonia volatilization loss rate during the rice season (%) |
|---|---|---|---|---|---|---|---|---|---|---|---|---|
| | | Ammonia emissions (kg ha$^{-1}$) | Proportion to total (%) | Loss rate (%) | Ammonia emissions (kg ha$^{-1}$) | Proportion to total (%) | Loss rate (%) | Ammonia emissions (kg ha$^{-1}$) | Proportion to total (%) | Loss rate (%) | | |
| 2021 | CK | 2.02e | 24.27e | – | 2.27c | 27.26ab | – | 4.03c | 48.47a | – | 8.32d | – |
| | WR | 30.49a | 34.94d | 37.64ab | 27.94a | 32.02a | 34.49a | 28.84a | 33.04b | 26.70a | 87.26a | 32.32a |
| | VvR | 24.51c | 48.73a | 37.83b | 11.19b | 22.24b | 17.26b | 14.61b | 29.03b | 16.91b | 50.31b | 23.29bc |
| | VsR | 27.38b | 47.40ab | 42.25a | 15.18b | 26.27ab | 23.42b | 15.21b | 26.33c | 17.60b | 57.76bc | 26.74ab |
| | RR | 16.90d | 40.15c | 26.07c | 11.41b | 27.11ab | 17.61b | 13.78b | 32.74b | 15.95b | 42.08c | 19.48c |
| | GR | 25.33bc | 43.08bc | 39.09a | 16.36b | 27.83ab | 25.25b | 17.10b | 29.09b | 19.80b | 58.80b | 27.22ab |
| 2022 | CK | 1.92d | 30.08c | – | 2.32d | 36.33a | – | 2.14c | 33.59a | – | 6.37d | – |
| | WR | 29.85a | 39.13b | 36.85a | 29.94a | 39.25a | 36.97a | 16.49a | 21.62b | 15.27a | 76.28a | 28.25a |
| | VvR | 21.97b | 42.16a | 33.90a | 18.72bc | 35.93a | 28.89bc | 11.42b | 21.91b | 13.22a | 52.11b | 24.13b |
| | VsR | 21.54b | 40.41ab | 33.24a | 19.94c | 37.42a | 30.78abc | 11.82b | 22.17b | 13.68a | 53.30b | 24.68ab |
| | RR | 16.47c | 38.26b | 25.41 b | 15.62bc | 36.31a | 24.11c | 10.94b | 25.43b | 12.67a | 43.03c | 19.92c |
| | GR | 22.58b | 39.21b | 34.85a | 22.72b | 39.45a | 35.06ab | 12.29b | 21.34b | 14.22a | 57.59b | 26.66ab |

**Table 4   Effects of different rotation patterns on rice yield traits.** Values within a column followed by different lowercase letters are significantly different at a significance level of 0.05. ns: not significant. * $P < 0.05$; ** $P < 0.01$. Abbreviations are the same as those in Table 2.

| Year | Treatment | Panicles per ($\times 10^4 ha^{-1}$) | Filled grains rate (%) | Spikelets per panicle | A thousand grain weight (g) | Yield of grain (t ha$^{-1}$) | Straw of grain (t ha$^{-1}$) |
|---|---|---|---|---|---|---|---|
| 2021 | CK | 284.52e | 93.26ab | 116.16b | 26.12a | 6.46c | 7.50b |
|  | WR | 334.12d | 90.34c | 125.10a | 25.75a | 9.91b | 10.29a |
|  | VvR | 359.03bc | 93.48ab | 125.47a | 26.06a | 10.06b | 10.95a |
|  | VsR | 379.50ab | 92.37b | 126.06a | 25.87a | 10.39a | 10.35a |
|  | RR | 350.17cd | 94.36ab | 125.34a | 26.33a | 9.94b | 11.05a |
|  | GR | 385.51ab | 94.68a | 126.33a | 25.98a | 10.71a | 11.21a |
| 2022 | CK | 286.12c | 89.55ab | 111.2b | 27.32a | 6.69c | 7.75d |
|  | WR | 337.57b | 85.11b | 125.83a | 25.52b | 9.80b | 10.02c |
|  | VvR | 348.31ab | 88.76ab | 126.99a | 26.60ab | 10.01ab | 11.65ab |
|  | VsR | 357.07ab | 89.15ab | 127.16a | 25.92ab | 10.07ab | 11.75ab |
|  | RR | 353.57ab | 90.64a | 126.51a | 25.60b | 9.96b | 10.68bc |
|  | GR | 364.29a | 89.12ab | 127.31a | 26.31ab | 10.66a | 12.52a |
| Variance analysis | Year | ns | ** | ns | ns | ** | ns |
|  | Treatment | ** | ns | ** | * | ** | ** |
|  | Year $\times$ Treatment | ns | ns | ns | ns | ** | ** |

**Notes.**

Values within a column followed by different lowercase letters are significantly different at a significance level of 0.05.

ns, not significant.

*$P < 0.05$

**$P < 0.01$

Abbreviations are the same as those in Table 2.

($P<0.05$), and increased by 3.08% to 8.07% compared with the other treatments. The yield of GR straw was 11.21 t hm$^{-2}$, which increased by 1.45% to 8.94% compared with the other nitrogen application treatments. There was no significant difference between the treatments. In 2022, the grain yield of the GR treatment was 10.66 t hm$^{-2}$, which increased by 5.86% to 8.78% compared with other nitrogen application treatments, significantly higher than that of the RR and WR treatments ($P<0.05$). The straw yield of the GR treatment was 12.52 t ha$^{-1}$, which was 6.55% to 24.95% higher than that of the other treatments and significantly higher than that of WR and RR ($P<0.05$); and there was no significant difference compared to VvR and VsR. Compared with WR, the combined application treatment increased the yield and yield traits of rice grains and straw.

The intensity of ammonia volatilization emissions is an important parameter for balancing rice yields and paddy field environments. As shown in Fig. 3, there was a significant difference in the intensity of ammonia volatilization emissions among the different treatments ($P<0.05$). In 2021, the ammonia volatilization emission intensities of rice fields treated with WR, VvR, VsR, RR, and GR were 8.81, 4.99, 5.56, 4.23, and 5.51 kg t$^{-1}$, respectively. WR was the highest and significantly higher than that of the other nitrogen application treatments ($P<0.05$). There were no significant differences in VvR, VsR, and GR. However, the intensity of ammonia volatilization emission in the VsR and GR treatments was significantly higher than that in the RR treatment ($P<0.05$). In 2022, the ammonia volatilization emission intensities of rice fields treated with WR, VvR, VsR,

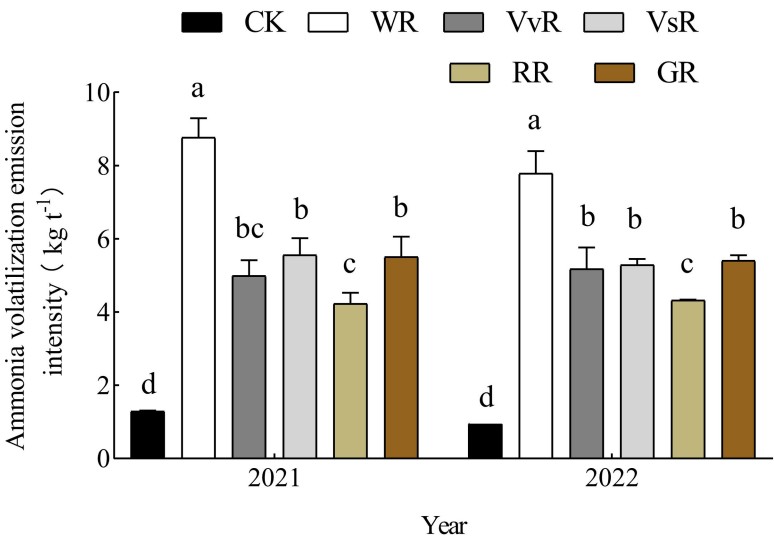

**Figure 3** **Effect of different treatments on ammonia volatilization emission intensity.** Different lower-case letters indicate significant differences between the treatments ($P < 0.05$).

RR, and GR were 7.78, 5.18, 5.29, 4.32, and 5.41 kg t$^{-1}$, respectively. WR was the highest and significantly higher than that of the other nitrogen application treatments ($P<0.05$). There were no significant differences in VvR, VsR, and GR but they were significantly higher than RR.

## SPAD of rice leaves

According to the determination of chlorophyll content (SPAD) in rice leaves during the panicle stage (Fig. 4), there were no significant differences in SPAD performance among treatments and years. In 2021, the difference in SPAD of rice leaves among the different treatments was GR>VvR>RR>WR>VsR, ranging from 42.99 to 43.42. The SPAD of VvR, VsR, RR, and GR rice leaves increased by 0.67%, 0.84%, 0.04%, and 1.01%, respectively, compared with the WR treatment. In 2022, the differences in SPAD of rice leaves among the different treatments were GR>VvR>VsR>RR>WR, ranging from 42.74 to 43.00. The average SPAD of rice treated with VvR, VsR, RR, and GR increased by 0.72%, 0.89%, 0.13%, and 1.37%, respectively, compared with the WR treatment.

## Effects on nitrogen uptake and utilization in rice

ANOVA revealed that the interaction of year, year, and treatment had no significant effect on grain nitrogen content, straw nitrogen content, nitrogen utilization efficiency, and nitrogen uptake (Table 5). The effect of treatment on grain nitrogen content, straw nitrogen content, nitrogen utilization efficiency, and nitrogen uptake was significant ($P<0.05$). From Table 5, it can be seen that the grain nitrogen content, straw nitrogen content, plant nitrogen uptake, and nitrogen utilization efficiency were the highest in GR and lowest in WR. Among them, the average nitrogen content of VvR, VsR, RR, and GR rice grains and straw increased by 0.33% to 6.99% compared with WR in the two years; 0.31% to 10.22%. Compared with WR, the two-year trials of VvR, VsR, RR, and GR

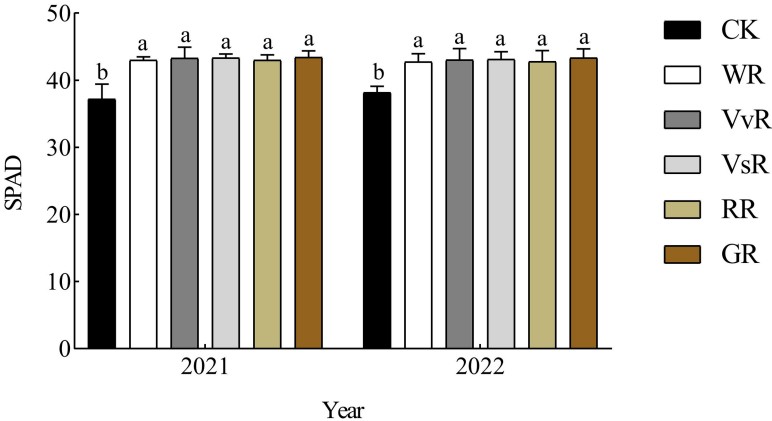

**Figure 4  Effects of different treatments on SPAD rice.** Different lowercase letters indicate significant differences between the treatments ($P < 0.05$).

**Table 5  Effects of different treatments on nitrogen content and utilization efficiency of rice.**

| Year | Treatment | Grain nutrient content (g kg⁻¹) | Straw nutrient content (g kg⁻¹) | Plant nitrogen absorption (kg ha⁻¹) | Nitrogen fertilizer utilization rate (%) |
|---|---|---|---|---|---|
| 2021 | CK | 11.46 c | 9.91 c | 148.07 d | – |
| | WR | 12.21 b | 11.39 b | 238.2 c | 33.38 d |
| | VvR | 12.85 ab | 11.53 ab | 255.52 b | 49.75 b |
| | VsR | 13.12 a | 11.62 ab | 256.58 b | 50.24 b |
| | RR | 12.25 b | 11.42 b | 247.9 c | 46.22 c |
| | GR | 13.32 a | 12.53 b | 283.12 a | 62.52 a |
| 2022 | CK | 11.46 c | 9.92 c | 153.53 d | – |
| | WR | 12.87 b | 11.74 b | 243.76 c | 33.42 d |
| | VvR | 13.09 ab | 11.94 ab | 270.14 b | 53.99 b |
| | VsR | 13.59 a | 11.85 ab | 276.05 b | 56.72 b |
| | RR | 12.92 b | 11.78 b | 254.50 c | 46.75 c |
| | GR | 13.50 a | 12.96 b | 306.22 a | 70.69 a |
| Variance analysis | Y | ns | ns | ns | ns |
| | T | * | ** | ** | * |
| | Y × T | ns | ns | ns | ns |

**Notes.**

Values within a column followed by different lowercase letters are significantly different at a significance level of 0.05. "-" means no data.

ns, not significant.

*$P < 0.05$

**$P < 0.01$

Abbreviations are the same as those in Table 2.

increased nitrogen utilization efficiency by an average of 18.47%, 20.08%, 13.08%, and 33.21%, respectively. Additionally, they increased the nitrogen uptake of rice by 9.05%, 10.48%, 4.24%, and 22.24%, respectively. Compared with conventional fertilization in rice, the green manure can effectively promote nitrogen uptake and utilization in rice.

## DISCUSSION

### Impact on ammonia volatilization in rice fields

Chemical fertilizers are considered to be the most important factors affecting ammonia volatilization (*Abulaiti et al., 2023*). This study found through two-year experiments that a 20% reduction in nitrogen treatment led to a decrease in ammonia volatilization emissions by 23.58 by 39.21 kg hm$^{-2}$ compared with conventional fertilization. This indicates that reducing nitrogen fertilizer application can effectively reduce ammonia volatilization in farmlands. This result is consistent with that of a previous study (*Yang et al., 2020*). In addition, climatic factors, particularly rainfall, also affect ammonia volatilization emissions from rice fields. A previous study suggested that in rainy weather after fertilization, the loss of ammonia volatilization in farmland is often lower, which may be related to the runoff caused by rainfall and the removal of some fertilizers (*Zheng et al., 2023*). The results of this experiment show that encountering rainwater during the middle and later stages of monitoring can lead to an increase in ammonia volatilization emissions the next day, forming a peak. It is speculated that this may be related to the presence of certain nitrogen deposits in the rainfall, and further analysis of the nitrogen content in rainwater is required.

Milk vetch, a common leguminous plant used as green fertilizer in winter fallow fields in southern China, has been a subject of study for many researchers investigating the effects of nitrogen on rice fields. *Xie et al. (2017)* studied a single-season rice milk vetch cloud planting model in southern China and found that it reduced ammonia volatilization accumulation by 14.6% compared with the urea single application treatment. Given the gradual loss of varieties of Chinese milk vetch and the significant reduction in cultivation area (*Xie et al., 2018*), it is particularly important to prioritize research on different types of green fertilizers. *Li et al. (2022)* found that applying different amounts of chemical fertilizers to hairy-leaved sweet potatoes reduces fertilizer usage by rolling green manure, with a 15% and 30% reduction in nitrogen application. Rolling green manure reduced ammonia volatilization emissions from rice fields by 5.45% and 9.97%, respectively. Under the same nitrogen application rate, there was no significant difference in ammonia volatilization emissions. Similarly, *Zhang et al. (2013)* used the green fertilizer of the February orchid (*Orychophragmus violaceus* L.) as a winter cover crop and constructed a green fertilizer spring corn rotation model. They found that reducing nitrogen application by 15% and 30% and turning over green fertilizers reduced ammonia volatilization emissions by 15.57% and 31.57%, respectively. However, under the same nitrogen application rate, the February orchid had increased ammonia volatilization emissions. It is speculated that this may be due to the short time for February orchids to be turned over and returned to the field and the low soil temperature, which leads to the insufficient decomposition of green manure. This study found that, based on a 20% reduction in chemical fertilizer, the ammonia volatilization emissions from rice fields were reduced by 37.02% and 31.97% on average over two years compared with conventional fertilization for both villous and vetch peas. This result is consistent with the previous conclusions. In recent years, rapeseed seeds have been rapidly developed as green fertilizer because of their low planting cost, high biomass, good fertilizer efficiency, short growth period, wide planting area, and strong adaptability

(*Zhang et al., 2020*). This study, based on a 20% reduction in chemical fertilizer, found that the average two-year reduction in ammonia volatilization emissions from subsequent crops compared with conventional fertilization was 47.68%, which was more significant than other treatments in reducing ammonia volatilization emissions. It is speculated that this is because rapeseed seeds are cruciferous crops and do not have the nitrogen fixation ability of leguminous crops, resulting in low nitrogen content in the plant itself. This can explain why the ammonia volatilization emissions of the rapeseed seed-rice treatment were lower than those of the leguminous green manure rice and wheat rice treatments and significantly lower than those of the purple cloud rice and wheat rice treatments. However, *Liu et al. (2022)* found that rapeseed-rice rotation had no significant effect on $NH_3$ emissions in a localized field experiment that began in 2014 in Central China, compared with WR. This may be related to the period when rapeseed green manure seeds were returned to the field. We used rapeseed as green manure and returned it to the field during the peak flowering period. The returning strain of green manure to the field can enhance soil organic matter. Organic matter composition can generate a higher amount of organic acids than rapeseed straw during the harvest period. The humus formed by it can enhance the ability of the soil to adsorb $NH_4^+$ while inhibiting soil ammonia volatilization (*Liang et al., 2022*).

## Impact on rice yield

Nitrogen fertilizer is an important agricultural method for promoting rice growth and increasing yield. However, excessive application of nitrogen fertilizer not only fails to increase yield but also results in crop yield reduction, resource wastage, and impacts farmland ecology. Green manure, as a high-quality and environment-friendly organic fertilizer source, planting green manure crops can effectively utilize light, temperature, water, nutrients, and land resources. The partial substitution of fertilizers is a reasonable and scientific fertilizer operation (*Zechmeister-Boltenstern et al., 2015*). Studies have been conducted on the impact of green manure combined with chemical fertilizers on crop yields (*Liang et al., 2022*; *Zhang et al., 2020*). The combination of Chinese milk vetch and nitrogen fertilizer can increase rice yield, even by reducing nitrogen fertilizer by 20% to 40%, which can ensure a stable yield of subsequent crops (*Zheng et al., 2022*). *Yang et al. (2020)* reported that the application of green manure with different gradients of reduced nitrogen fertilizer is more effective in increasing the yield of subsequent crops than conventional fertilization. The results of this experiment showed that by reducing the application of chemical fertilizer by 20%, the four kinds of green manure increased yield by 0.03 to 0.79 t hm$^{-1}$ compared with conventional rice wheat in the first year of fertilization, which is consistent with previous findings. Among the different fertilization treatments, Chinese milk vetch had the greatest impact on rice yield, which was significantly higher than other treatments, except for VsR ($P<0.05$), and increased yield by 2.98% to 8.07% compared with other treatments. This may be related to the return amount and nitrogen content of different green manure crops.

As a rich natural resource, leguminous green manure contains rhizobia in its root system, which can fix atmospheric nitrogen (*Xie et al., 2022*). Although rapeseed seeds are not leguminous, their high biomass and straw return can provide organic matter

and nutrients, thereby greatly increasing the yield potential of rice (*Wang et al., 2023a*; *Wang et al., 2023b*; *Wang et al., 2023c*). The results of this experiment also showed that the effect of applying green manure in the second year on rice yield was greater than that in the first year. This indicates that the combination of green manure and chemical fertilizer forms a sustainable fertilization model to achieve high crop yields. *Qiao et al. (2021)* found that combining rice straw and green manure can enhance yield components, leading to a significant increase in effective panicles. In this experiment, the yield traits of rice were improved under the combined application treatment compared with the conventional treatments, showing the highest increase in effective panicles, ranging from 4.77% to 11.65%. Except for the RR, all other treatments were significantly higher than WR ($P$<0.05), which is consistent with previous studies (*Zhang et al., 2020*). *Ma et al. (2017)* reported that the combination of green manure and nitrogen fertilizer can increase yield traits, such as grain per panicle and thousand-grain weight, thereby increasing rice yield. The results of this experiment showed that compared with the WR, the rice seed setting rate of each green manure-rice treatment increased by 3.88% to 5.47% in the two-year experiment, with significant differences between the treatments. The increase in grain number per spike was 0.37–1.08%, and the increase in thousand-grain weight was 1.02–2.72%, consistent with previous research. This may be due to the release of fertilizer efficiency by turning over and returning green manure, improving soil physical and chemical properties, especially nitrogen content, promoting rice growth, and enhancing the formation of effective panicles, which is conducive to establishing a scientific and reasonable population structure and laying a solid foundation for rice yield increases. The chlorophyll content (SPAD) is closely related to the intensity of leaf photosynthesis. *Islam et al. (2019)* found that compared with the application of chemical fertilizers alone, the combination of green manure and chemical fertilizers effectively increased the SPAD value of subsequent rice leaves. In this experiment, the SPAD values of rice under different green fertilizer treatments were higher than those under conventional rice and wheat treatments, with an increase of 0.08–1.19%. This indicates that turning green fertilizer increases the chlorophyll content of subsequent rice leaves, which is beneficial for delaying the decline in the leaf photosynthetic rate in the later stage of rice growth.

## Impact on nitrogen uptake and utilization efficiency in rice

Nitrogen is an important nutrient for rice growth and development. Blindly reducing nitrogen fertilizer application can lead to an inadequate soil nitrogen supply, affecting crop nitrogen uptake and utilization, ultimately limiting crop growth, and resulting in reduced yields (*Wang et al., 2023a*; *Wang et al., 2023b*; *Wang et al., 2023c*). Green manure is an important source of organic fertilizer that can replace other fertilizers, enhance soil fertility, and promote the uptake and utilization of nitrogen by subsequent crops (*Wang et al., 2022a*; *Wang et al., 2022b*). In particular, it can promote the distribution and transportation of nitrogen absorbed into grains during rice ripening and filling (*Zhang et al., 2013*). *Ma et al. (2017)* found that the combination of Chinese milk vetch and nitrogen fertilizer optimized the soil carbon-nitrogen ratio, promoted the mineralization of most organic matter into inorganic matter, and maintained a strong assimilation ability in the

middle and late stages of rice growth, which is conducive to nitrogen accumulation in the grain bank. *Yue et al. (2023)* conducted a field positioning experiment to study the effect of the combination of Chinese milk vetch and nitrogen fertilizer on nitrogen utilization efficiency. They found that nitrogen accumulation in rice plants treated with a combination of Chinese milk vetch and nitrogen fertilizer was higher than that in plants treated with nitrogen fertilizer alone. In this experiment, compared with the rice-wheat treatment, the combined application treatment in the two-year experiment increased rice nitrogen uptake by an average of 4.24–22.24%. Compared with conventional fertilization in WR treatments, the green manure exhibited a more effective promotion of nitrogen uptake and utilization in rice, which is consistent with previous findings (*Pu et al., 2023*; *Valizadeh et al., 2023*; *Liang et al., 2022*). This may be attributed to the synergistic effect of green manure and nitrogen fertilizer, which enhances the quality of the rice population, improves dry matter accumulation in rice, and ultimately increases nitrogen accumulation. Our results showed that different treatments resulted in significant differences in the ability of rice to absorb and utilize nitrogen. Returning green manure to the field can provide abundant carbon and nitrogen sources for soil microorganisms. This promotes the conversion of inorganic nitrogen to organic nitrogen in the soil, thereby reducing the levels of inorganic nitrogen in the soil and ensuring an adequate soil nutrient content. Variations in biomass and plant nitrogen content may contribute to differences in nitrogen uptake and utilization in rice under various treatments. Returning straw could potentially increase soil organic matter and total nitrogen as well as increase the soil microbial population and biomass. The diversity of microorganisms can further decompose green manure straw into nitrogen fertilizer forms that are more suitable for rice uptake and utilization (*Xie et al., 2022*). Improving the efficiency of nitrogen fertilizer utilization by crops is a major research focus in green rice production. Excessive nitrogen application hinders the uptake and utilization of nitrogen fertilizers by rice, leading to resource wastage and environmental pollution. In contrast, inadequate nitrogen application fails to meet the nutrient requirements of rice-growing areas (*Zhang et al., 2022*). The research results of *Zhang et al. (2022)* showed that Chinese milk vetch can replace a portion of nitrogen fertilizer, ensuring stable rice yield while improving nitrogen fertilizer utilization efficiency. The results of this experiment indicate that compared with the rice-wheat treatment, the average nitrogen utilization efficiency of each green fertilizer rice treatment in the two-year experiment increased by 13.08% to 33.21%. The nutrient content of different types of green fertilizers vary significantly, and their effects on rice growth, nutrient accumulation, and utilization also differ greatly. Chinese milk vetch, villous villosa, and vetch sativa are leguminous plants that develop root nodules. Therefore, leguminous green manure offers more advantages in nitrogen utilization and uptake owing to the biological nitrogen fixation effect of rhizobia than non-leguminous plants (such as rapeseed seed and wheat crops). Rapeseed seeds serve as an effective green fertilizer, offering significant biological benefits because of their high carbon and nitrogen ratios. The impact of green fertilizers on the growth, nutrient uptake, and utilization of subsequent crops, as well as on the reduction of ammonia volatilization and emissions, is complex. This study was based on a two-year field positioning experiment, and further research is required to obtain more accurate results.

## CONCLUSIONS

This study assessed the integrated influence of reducing nitrogen fertilizer and different green manures on ammonia volatilization loss in rice fields, agronomic traits, yield traits, nitrogen uptake and utilization, and rice yield. Our results showed that a 20% reduction in nitrogen fertilizer combined with various green manures significantly reduced $NH_3$ emissions and improved the apparent nitrogen utilization efficiency, leading to improved rice yield. Our results indicate that returning Chinese milk vetch as green manure is more beneficial for reducing ammonia volatilization in paddy fields and ensuring stable grain production in rice. Overall, this study presents an approach to reduce ammonia emissions and improve agronomic performance in environmentally friendly rice production based on a two-year field experiment. Nevertheless, further investigations under long-term stationary experimental conditions are required to gain a better understanding.

## ACKNOWLEDGEMENTS

The authors thank the editors and reviewers for their constructive comments and recommendations.

### Funding

This work was supported by the "Fengcheng Talent Introduction Plan" Youth Science and Technology Talent Support and Training Project of Taizhou (No. 2022) and by Special funds for the construction of national agricultural sustainable development pilot demonstration zones (No. TNY202205). The funders had no role in study design, data collection and analysis, decision to publish, or preparation of the manuscript.

### Grant Disclosures

The following grant information was disclosed by the authors:
Youth Science and Technology Talent Support and Training Project of Taizhou: 2022.
National agricultural sustainable development pilot demonstration zones: TNY202205.

### Competing Interests

The authors declare there are no competing interests.

### Author Contributions

- Zhongze Hu conceived and designed the experiments, performed the experiments, analyzed the data, prepared figures and/or tables, authored or reviewed drafts of the article, and approved the final draft.
- Daliu Yang performed the experiments, prepared figures and/or tables, and approved the final draft.
- Yaming Feng performed the experiments, authored or reviewed drafts of the article, and approved the final draft.
- Shuanglin Zhang performed the experiments, authored or reviewed drafts of the article, and approved the final draft.
- An Wang analyzed the data, prepared figures and/or tables, and approved the final draft.
- Qiaozhen Wang analyzed the data, prepared figures and/or tables, and approved the final draft.
- Yayun Yang analyzed the data, authored or reviewed drafts of the article, and approved the final draft.
- Chunying Chen analyzed the data, authored or reviewed drafts of the article, and approved the final draft.
- Yuefang Zhang conceived and designed the experiments, prepared figures and/or tables, authored or reviewed drafts of the article, and approved the final draft.
- Xian Wang conceived and designed the experiments, authored or reviewed drafts of the article, and approved the final draft.

## Data Availability

The raw measurements are available in the Supplementary File.

## Supplemental Information

Supplemental information for this article can be found online at http://dx.doi.org/10.7717/peerj.17761#supplemental-information.

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
