# Peer review of "Green manure combined with reduced nitrogen reduce NH3 emissions, improves yield and nitrogen use efficiencies of rice"

_PeerJ, doi:10.7717/peerj.17761_

## Round 0.1 · original submission · Major Revisions

The manuscript has initially reviewed by two experts and I received divergent comments: one recommended minor revision and another recommended rejection. I have then invited the 3rd reviewer, who suggested a major revision. I have read all three review comments and agree with most of them. Please modify your manuscript based on these very useful comments to improve the quality of the manuscript.

**Language Note:** The review process has identified that the English language must be improved. PeerJ can provide language editing services - please contact us at [email protected] for pricing (be sure to provide your manuscript number and title). Alternatively, you should make your own arrangements to improve the language quality and provide details in your response letter. – PeerJ Staff

Reviewer 1 ·

Basic reporting

The manuscript presents a valuable study on rice production with improving NUE. The language and logic should be improved to ensure that we can clearly understand your text, though the manuscript has been edited by a language edito.

Experimental design

Experimental design needs more detail. Was N fertilizer added in wheat or green manure seasons? Were all wheat and green manure returned into field?

Validity of the findings

no comment

Additional comments

1. The current title has an ambiguity, reduced N fertilizer?
2. Line 96-97, I think the research gap is wrong.
3. I cannot see your Tables.

Reviewer 2 ·

Basic reporting

By using a two-year field experiment, this study evaluated the integrated influence of nitrogen (N) fertilizer reduction and different green manures on ammonia (NH3) volatilization loss in rice fields, agronomic traits of rice, yield traits, N uptake and utilization as well as rice yield. The results are robust to show a 20% N fertilizer reduction with various green manures being possible for significantly reduced NH3 emissions and improved apparent N use efficiency and rice yield.

Experimental design

The two-year field experiment includes six treatments: no N input of rice-wheat roation, conventional N input (270 kg N/ha in rice season, the same below) of wheat-rice rotation, and reduced N input (216 kg N/ha) of four green manure-rice rotations. The experimental design is reasonable.

Validity of the findings

The main finding was that a 20% N fertilizer reduction with various green manures significantly reduced NH3 emissions and improved apparent N use efficiency and rice yield. Overall, the authors reported an approach to reduce NH3 losses and improve agronomic performance in environmentally friendly rice production through a two-year field experiment. However, further investigation under long-term stationary experimental conditions and various locations is required to understand the relationship deeply.

Additional comments

I suggest replace "nitrogen absorption" by "nitrogen (or N) uptake" through the whole text.

Reviewer 3 ·

Basic reporting

The English language should be improved to ensure that an international audience can clearly understand your text.
The latest references should be provided.

Experimental design

20% reduction in nitrogen fertilizer application in this investigation. Authors should provide the reference or previous experiment. And, the nutrient inputs during winter wheat and other green manure growing seasons should be listed.

Validity of the findings

the conclusion is not clear. Based on this two-year field investigation, which green manure application or crop rotation mode is more beneficial for ammonia volatilization reduction and/or stable grain production should be clarified.

Additional comments

Introduction: authors should revise this section briefly to highlight the key points, which should focus on the latest progress about green manure application all over the world, emission reduction measurements for ammonia volatilization, and the relationship between green manure, chemical fertilizer reduction, ammonia volatilization flux and intensity.
Materials and methods:
Line 107-108:Authors should provide the specific varieties and/or Latin name for wheat, rice, villous peas, vetch peas, rapeseed, and purple vetch in this investigation.
Line 115-116:villous villosa-rice (VvR), vetch sativa-rice (VsR), rapeseed-rice (RR), and milk vetch-rice (GR). These English name of green manure crops should be consistent with them in Line 107-108 and whole manuscript.
Line 117-118:WR treatment used 270 kg ha-1 nitrogen, and the other four treatments used 216 kg ha-1 nitrogen, 20% reduction in nitrogen fertilizer application in this investigation. Authors should provide the reference or previous experiment. And, the nutrient inputs during winter wheat and other green manure growing seasons should be listed.
Line 128-130: authors should provide the effective ingredient content (N, P2O5, K2O) of the applied fertilizers in this investigation.
Line 132-133: Ammonia volatilization Sampling and measurements method should be describe more detailedly, and the reference should be provided.
Line 140-141: chlorophyll meter SPAD-502 Plus, and Kaiser nitrogen method, authors should provide instrument model and/or manufacturer.
Line 152-168: in the section of Statistical count, the measurement unit for each parameter should be provided, for example, mg/kg, etc.
Results: authors should revise this section briefly to highlight the significant difference between these treatments and the annual differences.
Line 326: previous studies? Authors should provide supplementary references.
Line 337: single-season rice purple cloud planting model, authors should provide Latin name or scientific English name.
Line 339, purple cloud vetch; Line 351, Eryulan, authors should provide Latin name or scientific English name.
Discussion: authors should revise this section briefly to highlight the possible reason resulted in the significant difference for ammonia volatilization flux and rice grain yield, and nitrogen use efficiencies between these different experimental treatments, especially authors should discuss it with previous studies in paddy field investigation.
Conclusions: the conclusion is not clear. Based on this two-year field investigation, which green manure application or crop rotation mode is more beneficial for ammonia volatilization reduction and/or stable grain production should be clarified.
References: more references listed in this manuscript were chinese literature; And, according to the requirement of Peer J for reference format, authors should revise the references carefully.
Acknowledgements: authors should delete “The authors thank KetengEdit (www.Ketengedit.com) for English language editing during manuscript preparation.”

---

## Round 0.2 · Minor Revisions

Reviewer 1 has appreciated your efforts to address their comments and to improve the quality of your manuscript, but has a few more comments in your revised version.

Review 3 recommended an acceptance, but still has a number of suggestions to improve.

Please modify your manuscript accordingly.

Reviewer 1 ·

Basic reporting

The authors have addressed my concerns, as well as those of the other reviewers, and there is a notable improvement in the writing. However, the newly added content has introduced some new issues.
1. The authors have provided the basic physical and chemical properties of soil before the experiment for two years. Does this imply that the two-year experiments were independent? Additionally, the high total soil nitrogen levels might explain why there was no significant change in rice yield with reduced nitrogen treatments.
2. All treatments involved residue return, but the timing of residue return for green manure and wheat differs, leading to inconsistent aerobic decomposition of the straw. This needs to be clearly stated in the Materials and Methods section. Furthermore, the residual nutrient effects from wheat fertilization might impact ammonia volatilization during the rice season. The authors should address this issue.
3. Lastly, a major concern is that the study primarily presents observational results on ammonia volatilization and yield under different treatments, with potential mechanisms being discussed rather than empirically substantiated. For a submission to PeerJ, the data might appear somewhat insufficient and lack mechanistic support. Supplementing the study with data on soil nutrient dynamics and nitrogen cycling could significantly strengthen the manuscript.
In conclusion, while the manuscript has seen substantial improvements, addressing these new concerns through a major revision, especially by including more mechanistic data, would make it more robust and suitable for publication in PeerJ.

Experimental design

no comment

Validity of the findings

no comment

Reviewer 2 ·

Basic reporting

The revised article is written in clear English and has addressed all concerns raised by three reviewers. Now it can be acceptable for publication in this journal.

Experimental design

Experimental design is scientifically sound and the research questions are well defined.

Validity of the findings

The experimental data support the authors' findings in the article. The conclusions are well stated, which is linked to original research questions.

Additional comments

No additional comments.

Reviewer 3 ·

Basic reporting

Revision suggestions:
Line 5: check the author name.
Line 148-150: “The nitrogen, phosphorus, and potassium fertilizers used in the experiment were 46% urea, 52% superphosphate, and 12% potassium sulfate, respectively” should be revised as “The nitrogen, phosphorus, and potassium fertilizers used in the experiment were urea (N, 46%), superphosphate (P2O5, 52%), and potassium sulfate (K2O, 12% ), respectively.”
Line 183-200: authors can add this section in the “Statistical analysis”.
Line 443: nitrogen uptake might be more scientific.
Figure 1: Date in the figure 1 should be revised as Oct-2020, Jan-2021, etc.
Figure 2: check the information, some words are not clear.
Figure 3, 4: “Effect of different rotation patterns on ammonia volatilization emission intensity”, “Effects of different rotation patterns on SPAD rice” should be revised. The main topic of this manuscript is the comprehensive effect of green manure combined with reduced nitrogen, not different rotation patterns.

Experimental design

Experimental design is reasonable and scientific.

Validity of the findings

The result of this investigation is helpful to understand how to balance ammonia volatilization reduction and stable grain production in the paddy rice-winter wheat rotation field.

Additional comments

The authors have revised this manuscript carefully and completely throuth several weeks. I think it can be accept for publication in Peer J, but the authors need to double check and revise all the details of this manuscript, the revision suggestion are enclosed.

Annotated reviews are not available for download in order to protect the identity of reviewers who chose to remain anonymous.

---

## Round 0.3 · accepted · Accept

Thanks for your efforts to address review comments. I believe that all review comments have been address and this manuscript is ready for publication.